# The Postbiotic Properties of Butyrate in the Modulation of the Gut Microbiota: The Potential of Its Combination with Polyphenols and Dietary Fibers

**DOI:** 10.3390/ijms25136971

**Published:** 2024-06-26

**Authors:** Jessica Maiuolo, Rosa Maria Bulotta, Stefano Ruga, Saverio Nucera, Roberta Macrì, Federica Scarano, Francesca Oppedisano, Cristina Carresi, Micaela Gliozzi, Vincenzo Musolino, Rocco Mollace, Carolina Muscoli, Vincenzo Mollace

**Affiliations:** 1IRC-FSH Center, Department of Health Sciences, University “Magna Græcia” of Catanzaro, Germaneto, 88100 Catanzaro, Italy; rmbulotta@unicz.it (R.M.B.); rugast1@gmail.com (S.R.); saverio.nucera@hotmail.it (S.N.); robertamacri85@gmail.com (R.M.); federicascar87@gmail.com (F.S.); oppedisanof@libero.it (F.O.); carresi@unicz.it (C.C.); gliozzi@unicz.it (M.G.); v.musolino@unicz.it (V.M.); muscoli@unicz.it (C.M.); mollace@libero.it (V.M.); 2Department of Systems Medicine, University of Rome Tor Vergata, 00133 Roma, Italy; rocco.mollace@gmail.com

**Keywords:** short-chain fatty acids, butyrate, gut microbiota, Mediterranean diet, polyphenols, dietary fiber

## Abstract

The gut microbiota is a diverse bacterial community consisting of approximately 2000 species, predominantly from five phyla: Firmicutes, Bacteroidetes, Actinobacteria, Proteobacteria, and Verrucomicrobia. The microbiota’s bacterial species create distinct compounds that impact the host’s health, including well-known short-chain fatty acids. These are produced through the breakdown of dietary fibers and fermentation of undigested carbohydrates by the intestinal microbiota. The main short-chain fatty acids consist of acetate, propionate, and butyrate. The concentration of butyrate in mammalian intestines varies depending on the diet. Its main functions are use as an energy source, cell differentiation, reduction in the inflammatory process in the intestine, and defense against oxidative stress. It also plays an epigenetic role in histone deacetylases, thus helping to reduce the risk of colon cancer. Finally, butyrate affects the gut–brain axis by crossing the brain–blood barrier, making it crucial to determine the right concentrations for both local and peripheral effects. In recent years, there has been a significant amount of attention given to the role of dietary polyphenols and fibers in promoting human health. Polyphenols and dietary fibers both play crucial roles in protecting human health and can produce butyrate through gut microbiota fermentation. This paper aims to summarize information on the key summits related to the negative correlation between intestinal microbiota diversity and chronic diseases to guide future research on determining the specific activity of butyrate from polyphenols and dietary fibers that can carry out these vital functions.

## 1. Introduction

At birth, human organisms are colonized by a global microbiota consisting of bacterial species, viruses, and fungi. The microbiota inhabits various areas of the body, such as the skin, digestive tract, mouth, respiratory system, and more [1]. The largest and most famous microbiota is found in the intestines [2]: the microbiota in the gastrointestinal tract consists of approximately 100 trillion bacteria and 2000 bacterial species that change and adapt throughout life, becoming unique to each person [3]. Microbiota microorganisms outnumber somatic cells by a factor of 10 [4], and their genome is 150 times larger than the human genome [5,6]. The microbiota’s genetic information contains thousands of genes that are not found in the human genome. These genes play crucial roles in the host’s physiological functions [7]. The significance of the gut microbiota cannot be overstated, as it is anticipated to have crucial functions in preserving human health [8]. The microbiota bacteria and the host engage in a mutually beneficial symbiotic co-metabolism. Many factors contribute to the change in microbiota composition throughout life, such as nutrition, age, pH, lifestyle, and more [9,10]. In healthy adults, the intestinal microbiota is made up of five bacterial phyla: Firmicutes (79.4%), Bacteroidetes (16.9%), Actinobacteria (2.5%), Proteobacteria (1%), and Verrucomicrobia (0.1%) [11]. The dietary model suggests that the microbiota can be classified into three enterotypes, with each one dominated by a specific bacterial species. Enterotype 1, for example, is characterized by *Bacteroides*, which derives energy from fermenting proteins and carbohydrates. Enterotype 1 is linked to a typical Western diet, characterized by its high intake of animal proteins and fats and low fiber and vegetable content [12]. Enterotype 2 is associated with a fiber- and carbohydrate-rich diet, while Enterotype 3 relies on simple sugars for energy and may be connected to weight gain [13]. When the gut microbiota contains a large number and variety of microbial species, it creates a state of physical well-being called “eubiosis”. Conversely, intestinal “dysbiosis” can refer either to a general disruption of the normal bacterial flora or the overgrowth of harmful bacteria, leading to inflammation [14,15]. Intestinal dysbiosis is related to the occurrence of various diseases, such as obesity, hypertension, diabetes mellitus type 2, inflammatory bowel diseases, Crohn’s disease, ulcerative colitis, necrotizing enterocolitis, autoimmune diseases, and colorectal cancer [16,17]. Many scientific studies have shown a two-way relationship between the microbiota and the central nervous system, known as the gut–brain axis, which is now a major area of interest in neuroscience [18]. Various studies have indicated that interventions targeting the microbiota can improve symptoms and conditions of neurological diseases [19,20,21,22]. The microbiota’s bacterial species create numerous distinct metabolites that impact the host’s health. Among these molecules, short-chain fatty acids (SCFAs), products obtained by the digestion of food fibers and the fermentation of undigested carbohydrates by the intestinal microbiota, are well known. SCFAs are saturated fatty acids containing fewer than six carbon atoms. Typically, the colon contains these compounds: acetate, propionate, and butyrate [23]. Valerate, caproate, and formate are other SCFAs found in smaller quantities [24]. Colon cells absorb SCFAs through active transport mediated by monocarboxylate transporters (MCTs) and use them in the citric acid cycle to produce cellular energy [25]. Hepatocytes utilize non-metabolized SCFAs from the colon to generate energy and synthesize glucose, cholesterol, and fatty acids [26]. SCFAs that are left in small quantities enter the systemic circulation and peripheral tissues [27]. The whole metabolic process of SCFAs is represented in Figure 1. 

SCFAs have multiple beneficial effects on the epithelial, immune, nervous, and blood vessel systems [28]. A decrease in the production of these metabolites has been linked to various diseases such as intestinal inflammation, diabetes, liver cirrhosis, and atherosclerosis [29]. SCFAs play a crucial role in improving gastrointestinal health by acting locally on the intestine. These metabolites help to preserve the integrity of the intestinal barrier, aiding nutrient absorption and blocking pathogens and harmful substances [30,31]. Additionally, SCFAs enhance mucus production, which helps to lubricate intestinal epithelial cells and aids in digestion. Specifically, acetate and butyrate stimulate mucin secretion [32,33,34]. Research has shown that SCFAs can penetrate the blood–brain barrier (BBB) and access the brain. The three main SCFAs that accumulate in the brain follow this order: butyrate > propionate > acetate, at a ratio of 4.6:3.1:1.4 [35]. The passage of SCFAs through the BBB was verified through experiments involving radio-labeled SCFAs with _14_C [36,37]. The expression of SCFA receptors occurs in both the central and peripheral nervous systems [38], playing a crucial role in maintaining the integrity and functioning of the BBB. Close involvement with SCFAs is crucial for maintaining cerebral homeostasis and preventing the passage of toxins and pathogens into the brain due to BBB’s high selectivity [39,40]. Germ-free mice and mice treated with antibiotics [41,42] exhibit lower levels of various BBB junction proteins, including occludin, zonula occludens-1, and claudin-5 [43]. Certain fibers act as “prebiotics”, enhancing intestinal health and favoring beneficial microorganisms while preventing harmful ones. Additionally, certain types of food impact the microbiota by causing competitive interactions, affecting the development of SCAFs, regulating physiological changes, and enhancing the safeguarding of the mucous layer [44,45]. Inulin, galactose derivatives, fructo-oligosaccharides, gluco-oligosaccharides, -glucans, and lactulose are the most recognized prebiotics [46]. This paper aims to compile knowledge on butyrate to guide future research on determining the optimal levels of this metabolite from polyphenols and dietary fibers for its functional roles.

## 2. Role of Butyrate in the Intestine

Butyrate, propionate, and acetate are the main SCFAs produced in the gut microbiota. In the human large intestine, their concentration typically falls between 50 and 200 mM. The function of these compounds differs: butyrate serves as an energy source for the intestinal mucosa, acetate is closely involved in lipid synthesis (increasing the concentration of acetyl-Coa), regulating metabolic homeostasis, and propionate aids in gluconeogenesis in the liver [47]. SCFA production relies heavily on undigested carbohydrates, including non-starch polysaccharides, resistant starch, sugars, alcohols, non-digestible oligosaccharides, and proteins [48]. Butyrate, a crucial four-carbon SCFA, is found in varying concentrations in the intestines of mammals depending on their dietary fiber intake [49]. Butyrate is the preferred energy source for colon epithelial cells, but it also plays other important roles, such as modulating homeostasis and promoting cell differentiation and proliferation. It can reduce inflammation and oxidative damage and transform neoplastic cells while inhibiting histone deacetylases and binding to various G-protein-coupled receptors [50,51,52]. Carbohydrate glycolysis results in the formation of butyrate; colon cancer cells have a specific behavior that makes them use glucose instead of SCFAs as an energy source, leading to this protective effect. Due to this, butyrate can effectively carry out its histone modification function by inhibiting histone deacetylase and impacting the cell cycle. In contrast, intestinal epithelial cells utilize SCFAs as a source of energy, preventing butyrate from inhibiting histone deacetylase [53]. Cell cycle arrest caused by butyrate mainly happens in the G1 phase [54]. Moreover, cancer cell proliferation is counteracted by increased cell differentiation and cell cycle inhibition [55]. In addition, butyrate is responsible for activating pro-apoptotic genes like Bax and Bak while inhibiting anti-apoptotic genes such as Bcl-2 [56]. Several studies in the literature have found that butyrate improves the integrity of the intestinal barrier by regulating the expression of proteins in the tight junction [57].

The body activates the inflammatory process as a defense mechanism against harmful insults. However, the body has effective mechanisms to regulate inflammation, using self-limiting and negative feedback processes to stop pro-inflammatory signals and prevent prolonged damage [58]. The expression of pro-inflammatory cytokines, inflammatory enzymes, adhesion molecules, chemokines, and growth factors is regulated by NFκB [59]. Inflammatory bowel disease, autoimmune disease, and cancer can result from chronic inflammation in the intestine [60,61]. Numerous studies have shown that butyrate can inhibit NFkB [62] and activate PPARs [63]. The concentration of butyrate in the system is roughly a thousand times lower than in the intestine. It is crucial to recall that butyrate also functions in the gut–brain axis [64]. The intestine and brain communicate bidirectionally via the vagus nerve, neuroendocrine pathways, and neuroimmune pathways [65,66], which involve the central nervous system, enteric nervous system, and various neurons that connect the brain and intestines [67]. In this context, it is worth noting that butyrate can cross the BBB and activate the vagus nerve, thereby impacting the hypothalamus and influencing appetite and eating behavior [68]. Cholinergic enteric neurons can be increased through the epigenetic effect of butyrate [69].

### Butyrate Participates in Maintaining the Integrity of the Blood–Brain Barrier

A hypothesis has been proposed about the close relationship between the gut microbiota and the nervous system in recent decades [5]. Researchers have been fascinated by this connection, leading to an increase in knowledge about this topic [70,71]. Although not fully understood, intestinal microbial populations can produce neurotransmitters like serotonin and GABA, which have been shown to affect the nervous system by crossing the blood–brain barrier [72]. Thus, any approach that can impact the makeup of the gut microbiota, like nutrition, can have both positive and negative effects on brain disorders and significant diseases [73]. The BBB specializes in maintaining cerebral homeostasis, regulating ion concentration, protecting against brain toxins, supporting glial and neuronal activity, and defending against infections [5,74]. The BBB becomes more permeable when the barrier malfunctions, disrupting brain tissue homeostasis. Additionally, a higher permeability of the BBB leads to the entry of toxic molecules and the subsequent inflammatory process, both of which contribute to neurodegeneration [75]. Studies have demonstrated that germ-free mice have a higher blood–brain barrier permeability than mice with a healthy microbiota [76]. The loss of BBB integrity is the main cause of several neurological disorders, including Parkinson’s disease, Alzheimer’s disease, depression, epilepsy, multiple sclerosis, mental and behavioral damage, and autism spectrum disorder [77,78]. Moreover, when bacterial strains were reconstituted in animal microbiota, BBB permeability was maintained [79]. Previous studies have demonstrated the significant role of SCFAs, specifically butyrate, in the central nervous system. This includes preventing neuroinflammation, facilitating microglia maturation, aiding neurodevelopment, influencing neurotransmitters, and promoting neurogenesis [80,81]. To validate these discoveries, manipulating the gut microbiome through fecal microbiota transplantation and probiotic use demonstrated that butyrate is crucial for maintaining BBB integrity [82,83]. A reduction in plasma or fecal levels of butyrate can be considered as a biomarker of several neurological disorders [84] and has been observed in many pathological states, such as stroke, multiple sclerosis, vascular dementia, encephalopathy, and traumatic brain injury [85,86]. 

The rapid uptake of butyrate from plasma to the brain was observed in experiments. The mechanisms of action responsible for the action of butyrate in the health of the BBB are multiple. It promotes the secretion of mucin, which helps to reduce inflammation and the absorption of lipopolysaccharides. Butyrate also increases the integrity of the BBB by improving the activity of antioxidant systems and increasing the expression of tight junction proteins [87,88]. In the inflammatory process linked to intestinal dysbiosis, various inflammatory mediators cause increased BBB permeability by activating microglia and upregulating the expression of adhesion molecules and chemokine receptors [89]. By acting on the immune system, increased butyrate decreases systemic inflammation and suppresses the production of inflammatory mediators.

Figure 2 shows these properties explained for simplicity on the colonocytes; however, it should be remembered that butyrate also practices them specifically on some cell lines of the nervous system.

## 3. The Protective Role of Polyphenols and Dietary Fibers

The strong connection between lifestyle and health management is widely acknowledged and extensively documented in the scientific literature [90,91]. Overall, eating habits can be modified and have a significant impact on human physiology, health, and cognitive functions [92,93]. The Healthy Eating Index (HEI) was created to measure how well diets align with American dietary guidelines. Initially, the use of the HEI was primarily for scientific purposes to measure diet quality. However, today, the HEI is used to evaluate the adherence of any food group to key dietary recommendations [94]. Many dietary models have been created to address various health conditions, and one of the most prominent is the Mediterranean diet (MD), which is known for its beneficial effects in preventing chronic diseases. The MD is centered around consuming fresh fruits, vegetables, legumes, fiber, vitamins, unrefined cereals, and extra virgin olive oil, with moderate amounts of fish, dairy products, and ethanol (particularly red wine during meals), and a limited intake of red meat [95,96]. Alongside the MD, another well-known dietary model is the Dietary Approach to Stop Hypertension (DASH), which effectively lowers blood pressure. Typically, a complete and varied diet prevents deficiencies and maintains good health in the absence of other diseases. Polyphenols and dietary fiber, known for their numerous health benefits, are especially crucial among all foods. Food polyphenols, numbering around 10,000 compounds, are classified based on various characteristics, like origin, structure, and function. Polyphenols are classified into flavonoids (flavones, flavanones, flavonols, flavanols, isoflavones, and anthocyanins) and non-flavonoid molecules (stilbene, phenolic acids, tannins, lignans, and hydroxycinnamic acids) based on their structural characteristics [97]. Polyphenols offer numerous health benefits, safeguarding against chronic diseases and influencing physiological processes like enzymatic activity and cellular redox. The donation of electrons/hydrogens by polyphenols and the elimination of radical structures are attributed to the phenolic structure [98]. Additionally, polyphenols can inhibit NF-κB, a transcription factor that plays a role in inflammation, cell survival, and growth. NF-κB is also involved in the development of inflammatory bowel diseases and colorectal cancer while promoting antiproliferation and apoptosis [99]. Additionally, polyphenols regulate the activity of kinases like Akt/protein kinase, tyrosine kinase, and mitogen-activated protein kinase (MAPK) [100]. Polyphenols finally hinder certain pro-inflammatory enzymes, like 5-lipoxygenase and cyclooxygenase, which helps to prevent colorectal cancer [101]. The majority of polyphenols are typically found in their glycosylated forms but can undergo structural changes through esterification reactions [102]. Polyphenols have a low bioavailability due to factors like hepatic metabolic processes, interaction with the food matrix, and the action of the intestinal microbiota. Polyphenols are typically considered xenobiotic after being ingested, resulting in a significant decrease in their bioavailability. In fact, in vitro studies have shown that concentrations of 10 to 100 μM of polyphenols have exhibited anticancer or anti-inflammatory effects. Currently, it is understood that only a small percentage (5–10%) of polyphenols are absorbed in the small intestine, while the majority (90–95%) accumulate in the large intestine and undergo enzymatic processes by the gut microbiota [103]. Low-molecular-weight phenolic metabolites are formed through polyphenolic demolition reactions, which are attributed to the gut microbiota. Scientific evidence strongly suggests that polyphenol metabolites have a positive impact, even after undergoing biotransformation [104]. The bioavailability of polyphenols relies on their intake, size, and the composition of intestinal microbiota [105]. Recent research has demonstrated that the gut microbiota can metabolize polyphenols in food. Polyphenols metabolized by gut bacteria produce representative metabolites such as equol, urolithin, and esperitotin [84]. Multiple studies, conducted in animals and humans, indicate that specific amounts of polyphenols can alter the gut’s microbial makeup, either inhibiting or promoting the growth of certain groups [106]. For example, the consumption of polyphenols in wine has significantly increased the abundance of Bacteroides, Bifidobacterium, Enterococci, Prevotella, and Blautia coccoides-E in the human rectale group. The polyphenolic fraction seems to have both prebiotic and selective antimicrobial effects against intestinal pathogenic bacteria [107,108]. Several factors, such as polyphenol structure, dosage, and bacterial strain, influence how polyphenols affect growth and metabolism [109]. As an illustration, the flavonoid B-ring can easily insert itself between nucleic acid bases, disrupting DNA synthesis and RNA [110]. The interaction between polyphenols and gut bacteria impacts SCFA production, leading to a significant increase. In a controlled crossover study, researchers found that polyphenols from freeze-dried cranberry powder increased available butyrate levels by reducing its elimination with feces [111]. The interaction between polyphenols and microbiota has been shown through the incubation of polyphenols with fecal samples and in vivo due to polyphenolic food supplementation [112,113]. Polyphenols and the microbiota have a mutual interaction, allowing for a crosstalk between the two parties.

Fiber is an edible carbohydrate polymer with three or more monomeric units, bound by glycosidic bonds, and is not digested or absorbed in the human small intestine [114]. Currently, defining dietary fiber accurately is a complex task that involves considering various factors like size, solubility, fermentability, viscosity, composition, source, and more [115]. Dietary fiber can be categorized as soluble or insoluble. The absorption of water by insoluble fiber increases the specific weight and volume of feces, making them softer and promoting intestinal motility [116]. Cellulose is the most prevalent insoluble fiber found in nature, typically found alongside hemicellulose, lignin, and pectin. Water is highly attracted to soluble fiber and causes it to dissolve in a watery solution. Moreover, the majority of soluble dietary fibers ferment in water, causing them to swell and form a gel-like structure [117]. Soluble fibers, like pectins, mucilages, galactomannans, and gums, can be consumed through food or dietary supplements [118]. They are primarily present in plant-based foods and have several beneficial effects [119,120,121], including lowering the risk of gastrointestinal diseases like colorectal cancer and irritable bowel syndrome. Including soluble fiber is crucial for maintaining a balanced and healthy dietary model [122,123]. Soluble fiber also helps with controlling appetite, enhancing insulin sensitivity, and reducing weight. This particular fiber can slow down the breakdown and absorption of energy nutrients like starch and triglycerides. This leads to a lower overall energy intake, including glucose and cholesterol, which reduces the risk of type 2 diabetes, obesity, and metabolic diseases [124,125]. Despite the presence of both soluble and insoluble fibers in most plant-based foods, their consumption is notably low, especially in Western countries. Fortified foods were created to boost fiber intake by adding indigestible carbohydrates, polymers, and oligosaccharides to regular foods [126]. Epidemiological studies have shown that African people, who consume a diet rich in dietary fiber, have a lower incidence of colorectal cancer. Certain fibers have a “prebiotic” effect on the gut microbiota by selectively promoting beneficial bacterial populations [127]. SCFAs are formed through the fermentation of dietary fiber by the gut microbiota, as described previously. As a result, a diverse diet high in fiber promotes a healthier gut microbiota and increased production of SCFAs, thus maintaining intestinal health [128]. Conversely, a diet lacking in fiber but high in proteins and sugars can lead to reduced bacterial diversity in the microbiota, decreased SCFAs, and the development of chronic inflammatory diseases. To prevent infections and microbial invasions, the intestinal epithelium is covered and protected by a well-structured and compact mucus layer. Consuming a high-fiber diet promotes the creation of protective mucus that is stable. Animal model studies have shown that the amount of fiber used was higher than recommended for daily consumption. Instead of the recommended daily fiber intake of 30 g, approximately 100 g/day was consumed [129]. Based on these studies, it is reasonable to think that dietary fiber supplementation is inadequate. Currently, the recommended daily intake of dietary fiber for optimal health benefits is over 50 g [130]. Since it may be challenging to consume these quantities through regular food, it is recommended to consider taking supplements for adequate supplementation [131]. It would be intriguing to assess the outcomes of administering butyrate along with polyphenols or fibers, following the discussion on their impact on SCFA production. Positive and regenerative feedback is likely to lead to an increased availability of butyrate and improved intestinal health.

After describing the polyphenols and the dietary fiber and showing how their intake increases the production of SCFAs, it would be interesting to evaluate the results obtained from the administration of butyrate, which is a main ingredient in several dietary supplements (e.g., Colonzak), with the addition of polyphenols or fibers. It is likely to result in positive and regenerative feedback, responsible for a net increase in available butyrate and for achieving intestinal health. Figure 3 illustrates the effects of polyphenols (panel a) and fibers (panel b) on the gut microbiota.

## 4. Source of Butyrate

Since a greater production of butyrate determines, as already said, a protective effect on the whole organism, it is fundamental to know the main strategies involved in its production. An initial source of butyrate is undoubtedly nutrition: some foods are rich in this compound, including dairy products such as butter (3 g/100 g), goat’s cheese (1 to 1.8 g/100 g), and whole cow’s milk (0.1 g/100 g). However, it is impossible to consume excessive amounts of these foods in the daily diet in order to avoid the occurrence of other metabolic and cardiovascular disorders [144]. As a result, the safest and most immediate pathway for butyrate production remains the fermentation of appropriate foods by the gut microbiota. The amount of butyrate depends on a variety of factors: the quality and quantity of the microflora present in the colon, the type of substrate taken with feeding, and the time of its intestinal transit [145]. To produce butyrate, intestinal bacteria must have the appropriate enzymes to break down substrates. The majority belong to the phylum of the *Firmicutes* of the genus *Clostridium*, and, more in particular, to the family of the Ruminococcaceae and the Lachnospiraceae, such as Roseburia intestinalis, Faecalibacterium prausnitzii, Eubacterium rectale, E. Halli, and E. cylindroides [146]. During life, the microbiota is in a state of dynamic equilibrium: in fact, it changes and is very sensitive to dietary, physiological, and/or environmental changes; its composition varies in each individual according to factors such as the place where one lives, personal history, genetic heritage, lifestyle, feeding, and type of birth [73]. The butyrate molecule is produced by the fermentation of different substrates, such as resistant starches (polysaccharides present in whole grains, starchy foods, bananas, and potato starch) and beta-glucans (oats, barley, and rye) [147]. Finally, intestinal transit time is extremely important for the levels of butyrate: numerous studies have shown that a shorter gut transit time leads to reduced development of intestinal bacterial growth and that an insufficient microbiota slows down the production of butyrate [148].

Another strategy to increase the levels of butyrate in the body is to expand the bacteria that produce butyrate; this compound is produced not only in the human colon but also in non-human environments [149]. Many outdoor environments also have conditions that promote the growth of butyrate-producing bacteria, such as the presence of degradable organic materials within bulk soils, intestines of animal carcasses, or plants [150]. These butyrate-producing bacteria can be easily transported into the house, turning themselves from outdoor to indoor sources. Many Gram-positive anaerobic bacteria, producing butyrate, create endospores that allow for their survival in a dormant condition and aerobic outdoor situation. Subsequently, when they come into contact with the human intestine (indoor), they can trigger the germination of the spores, increasing taxa of the gut microbiota and the levels of butyrate [151]. In a randomized controlled mouse study, Liddicoat et al. [152] have demonstrated that butyrate-producing bacteria present in the soil are transferred by dust from the soil to the gut microbiota of mice, increasing the levels of butyrate in the animals. The increase in butyrate-producing bacteria from outside occurs through open doors and windows, clothing, shoes, and pets. In light of these considerations, we can say that the bacteria present in the air inside the house come mainly from a combination of human/pet activity and the outside air that arrives inside [153].

## 5. Conclusions

Butyrate is a short-chain fatty acid that mammals produce in their gut through the fermentation of dietary fiber by the microbiota. This molecule is versatile and protects against multiple diseases, including diabetes, intestinal inflammation, obesity, colon cancer, and neurological disorders [154,155]. Additionally, butyrate provides multiple approaches to maintain the BBB’s integrity. This review focused on three important topics: identifying the importance of intestinal microbiota and SCFA production; knowing the various roles of butyrate; and understanding the importance of consuming polyphenols and fiber in the diet. In light of the literature cited, butyrate could be considered as a postbiotic compound, able to regulate the immune system by exploiting its metabolic, anti-inflammatory, antioxidant, and antiproliferative properties; it could be indicated for the treatment of gastrointestinal and extra-intestinal diseases. However, to reach these conclusions, further research is needed: first, it would be essential to determine precisely how the bioavailability of butyrate varies after its production. Subsequently, the precise concentrations of butyrate produced after the ingestion of polyphenols and fibers should be identified to determine the exact amount of these foods to be consumed daily. Finally, clinical studies would be suggested to test the use of butyrate for its postbiotic activities.

## Figures and Tables

**Figure 1 ijms-25-06971-f001:**
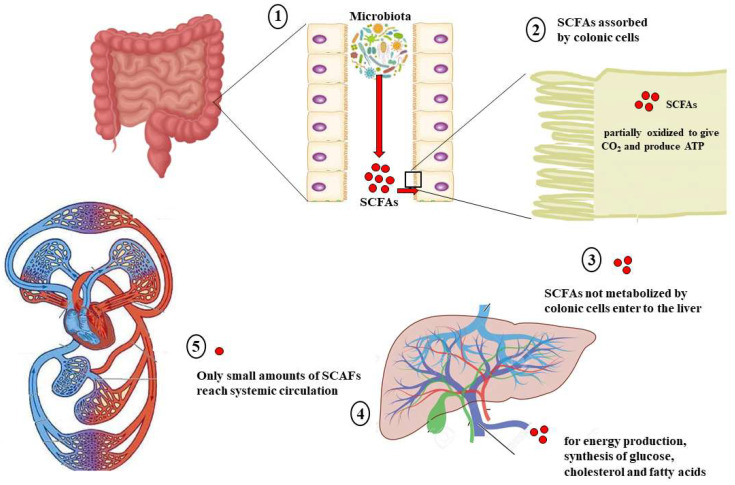
The fate of SCFAs. Following dietary intake of fiber, the digestive process begins in the intestine, (1)where the microbiota produces SCFAs. (2) Colonic cells quickly absorb these compounds and convert them into CO_2_ and ATP for energy. (3) The unmetabolized SCFA portion reaches the liver, (4) providing energy to hepatocytes and aiding in the synthesis of glucose, cholesterol, and fatty acids. (5) Finally, only a very small part of SCFAs reaches systemic circulation.

**Figure 2 ijms-25-06971-f002:**
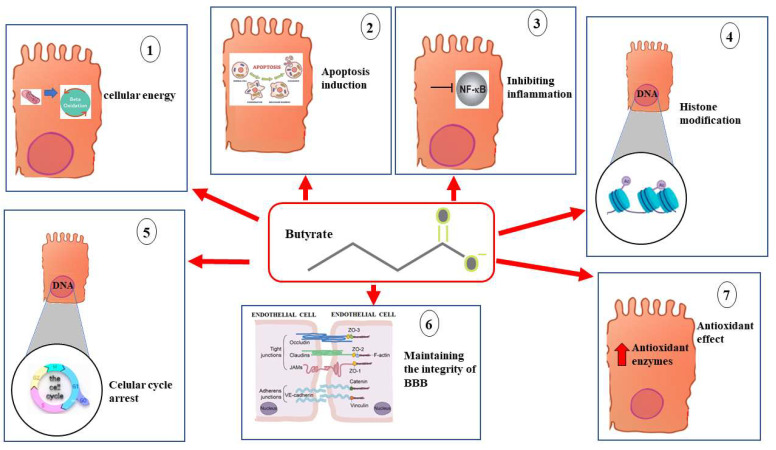
Main functions performed by butyrate on colonocytes. 1. Cellular energy; 2. apoptosis induction; 3. anti-inflammatory effect; 4. histone modification; 5. cellular cycle arrest; 6. maintaining the integrity of BBB; 7. antioxidant effect.

**Figure 3 ijms-25-06971-f003:**
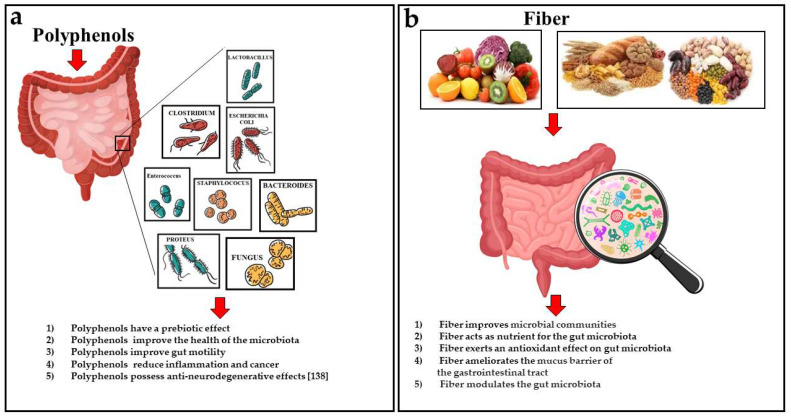
Panel (**a**) shows the main effects of polyphenols on the gut microbiota [132,133,134,135,136,137,138], while panel (**b**) highlights those exerted by fibers [139,140,141,142,143].

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
