# Peer review of "The Postbiotic Properties of Butyrate in the Modulation of the Gut Microbiota: The Potential of Its Combination with Polyphenols and Dietary Fibers"

_ijms, 2024, doi:10.3390/ijms25136971_

Round 1
Reviewer 1 Report
Comments and Suggestions for Authors
The paper is well written and interesting. You could include a paragraph on strategies to deliver and increase butyrate levels and butyrate producing bacteria.
Author Response
Dear reviewer,
thanks for your advice. I send you attached my responses to your comments, which appear (even in the manscript) highlighted in blue.
Regards,
Jessica

Reviewer 2 Report
Comments and Suggestions for Authors
1. In Figure 1, consider omitting the numbers one and two. You may begin with number three, indicating that the gut microbiota ferments fiber and carbohydrates into SCFAs. In the figure caption, provide a detailed description of each numbered process depicted in the figure.
2. In line 115, the phrase “acetate increases systemic concentrations in the blood” is vague. Please specify the substances whose concentrations are increased by acetate.
3. In Figure 2, all cells appear similar, but some are labeled as “colonocyte.” Are all the cells depicted intended to be colonocytes? According to the text, butyrate functions in various tissues and cells. Please clarify this distinction within the figure.
4. In Figure 3, include the effects of phenols, polyphenols, and fiber on the gut microbiota, supported by relevant literature. Ensure that this information is also reflected in the figure caption.
5. Before the conclusion section, it is necessary to outline directions for future research that address the points raised in the conclusions. Specify the types of studies required to investigate the primary questions regarding the role and effects of butyrate as a potential postbiotic.
6. In the conclusions section, avoid listing the relevant points numerically. Rephrase the conclusions to present the same information in a more fluid and integrated prose style.
Author Response
Dear reviewer,
thanks for your advice. I send you attached my responses to your comments, which appear (even in the manscript) highlighted in green.
Regards,
Jessica
